# Microplankton life histories revealed by holographic microscopy and deep learning

Harshith Bachimanchi[1], Benjamin Midtvedt[1], Daniel Midtvedt[1], Erik Selander[2], Giovanni Volpe[1]*

[1]Department of Physics, University of Gothenburg, Gothenburg, Sweden; [2]Department of Marine Sciences, University of Gothenburg, Gothenburg, Sweden

*For correspondence: giovanni.volpe@physics.gu.se

Competing interest: The authors declare that no competing interests exist.

**Abstract** The marine microbial food web plays a central role in the global carbon cycle. However, our mechanistic understanding of the ocean is biased toward its larger constituents, while rates and biomass fluxes in the microbial food web are mainly inferred from indirect measurements and ensemble averages. Yet, resolution at the level of the individual microplankton is required to advance our understanding of the microbial food web. Here, we demonstrate that, by combining holographic microscopy with deep learning, we can follow microplanktons throughout their lifespan, continuously measuring their three-dimensional position and dry mass. The deep-learning algorithms circumvent the computationally intensive processing of holographic data and allow rapid measurements over extended time periods. This permits us to reliably estimate growth rates, both in terms of dry mass increase and cell divisions, as well as to measure trophic interactions between species such as predation events. The individual resolution provides information about selectivity, individual feeding rates, and handling times for individual microplanktons. The method is particularly useful to detail the rates and routes of organic matter transfer in micro-zooplankton, the most important and least known group of primary consumers in the oceans. Studying individual interactions in idealized small systems provides insights that help us understand microbial food webs and ultimately larger-scale processes. We exemplify this by detailed descriptions of micro-zooplankton feeding events, cell divisions, and long-term monitoring of single cells from division to division.

## Editor's evaluation

This paper presents a valuable new method combining holographic microscopy and deep learning to track the behavior and growth of individual plankton. The paper illustrates the method with compelling data from two applications, zooplankton feeding behavior and diatom cell division. This paper will be of interest to plankton ecologists and ocean ecosystem modelers. The results obtained from this method will provide new insights into the trophic strategies of ocean plankton and important constraints for global ocean models.

## Introduction

The role of herbivores in structuring plant communities is well established in terrestrial ecology. Already Darwin, in his foundations on evolutionary biology (**Darwin, 2004**), noted how excluding herbivores from a heath land transformed it into a forest of pine trees with an altogether different species composition. Single-celled micro-zooplankton take on the role of herbivores in the ocean, consuming approximately two thirds (40 Petagrams (Pg) carbon) of the primary production (**Calbet and Landry, 2004**). In oceanic ecology, the primary production is dominated by unicellular phytoplankton,

**eLife digest** Picture a glass of seawater. It looks clear and empty, but in reality, it contains one hundred million bacteria, about one hundred thousand other single-celled organisms, and a few microscopic animals. In fact, the majority of life in the ocean is microscopic and we know relatively little about it. Nevertheless, these microbes have a major impact on our lives. Microscopic algae known as phytoplankton, for example, produce half of the oxygen we breathe.

For animals, birds and other large organisms in the ocean, we have a good understanding of who eats who and where the material ends up. However, for phytoplankton and other microbes, we depend on bulk measurements and averages of large groups. Bachimanchi et al. developed a method to follow individual microbes living in seawater and to observe how they move, grow, consume each other and reproduce.

The team combined holographic microscopy with artificial intelligence to follow multiple planktons, diatoms and other microbes throughout their life span and continuously measured their three-dimensional location and mass. This made it possible to estimate how fast the organisms were growing and moving, and to observe what they ate. The experiments revealed new insights into how micro-zooplankton, diatoms and other microbes in the ocean interact with each other.

This new method may be useful for researchers who would like to track the movements and whereabouts of microscopic planktons, bacteria or other microbes for extended periods of time. It is also a rapid method for counting, sizing, and weighing cells in suspension. The hardware used in this method is relatively cheap, and Bachimanchi et al. have shared all the computer code with examples and demonstrations in a public database to enable other researchers to use it.

which produce around 50 Pg of carbon annually, quantitatively slightly exceeding the production of terrestrial plants (*Behrenfeld and Falkowski, 1997*; *Field et al., 1998*). Selective grazing shapes the plankton community and drives large-scale processes such as harmful algal bloom formation and carbon export (*Irigoien et al., 2005*; *Selander et al., 2019*).

Despite its importance, our understanding of the role of micro-zooplankton in shaping oceanic communities is still much less developed than that of macro-organisms, which can more readily be observed at the individual level (*Glibert and Mitra, 2022*). In fact, rates and fluxes in the oceanic microbial food web are still mainly inferred from indirect measurements or ensemble averages, leaving us with a limited mechanistic understanding. Quantitative estimates of primary production are mostly inferred from satellite images of ocean color (chlorophyll) using moderate resolution spectroradiometers calibrated against in situ isotope incorporation experiments (*Hu et al., 2012*). Ensemble-level biomass transitions during grazing events by microscopic zooplankton are calculated from dilution experiments (*Landry and Hassett, 1982*), where the grazer density is manipulated by dilution, and the corresponding net increase in primary production is approximated. While these methods provide good estimates of the magnitude of biomass fluxes, they do not resolve the small-scale individual interactions that drive the large-scale processes. Moreover, indirect measurements of processes such as micro-zooplankton grazing rest on assumptions that are not always fulfilled. For example, feeding rates and growth rates of both predators and prey need to be unaffected by dilution, which is often not true (*Dolan et al., 2000*). In addition, the dilution technique is based on chlorophyll measurements and does not account for consumption of non-chlorophyll-bearing particles, which leads to underestimation of carbon transfer (*Stoecker et al., 2017*).

Currently, the biomass of individuals is often inferred from volume-to-carbon relationships developed over time for different trophic groups of planktons (*Strathmann, 1967*; *Menden-Deuer and Lessard, 2000*), which require cell counting and sizing followed by elemental analysis, but do not allow continuous measurements of the same individual. However, these regression relations are not very precise: the average deviation of individual data points to the regressed expression exceeds 50% (*Menden-Deuer and Lessard, 2000*). In addition, single cells of the same volume can differ by a factor two in dry mass, which is not possible to detect by volume-to-carbon relationships. To go beyond the current level of detail in marine microbial food webs, we need complementary techniques that can follow individual microplanktons over extended periods, while continuously monitoring their growth rate and predation events.

Continuous measurements can be realized using microscopy techniques. For example, holographic microscopy can record holograms of cells under investigation in the form of interference patterns containing phase and amplitude information. The information in the holograms can be used to extract the three-dimensional position of microplanktons as well as their mass (*Zangle and Teitell, 2014*). Holographic imaging has already found applications in microbial studies, especially for in situ measurements of particle size distributions and their identity (*Nayak et al., 2021*). However, its full potential has not yet been exploited, namely for the quantitative investigation of the growth and feeding patterns of individual planktons over prolonged times. Arguably, this is because the data acquisition and processing pipelines are very computationally expensive.

Here, we solve this problem by employing a technique that combines holography with deep learning. The deep-learning algorithms circumvent the long computational times and, once trained, allow rapid determination of three-dimensional position and dry mass of individual microplanktons over extended time periods. We evaluate this method on nine plankton species belonging to different trophic levels and representing the major classes of microplankton. We highlight that unlike other methods, our approach makes it possible to follow and weigh single cells throughout their lifetime, being especially useful to detail micro-zooplankton and mixotrophic life histories as feeding events can be quantitatively measured. Furthermore, the estimated dry mass can be tagged to single planktons detected in the experiments. We can track and identify both prey and predator cells and closely follow the transfer of mass from cell to cell. Finally, we observe the growth and cell divisions in diatoms by their long-term monitoring over more than one cell cycle.

## Results

### Experimental setup and deep-learning data analysis

*Figure 1* shows an overview of the holographic microscopy experimental setup and the deep-learning data analysis pipeline to estimate the position and dry mass of the planktons. We use an inline holographic microscope in a lens-less configuration (see details in Methods, 'Holographic imaging'). A monochromatic LED light source illuminates the sample suspension that contains the planktons under investigation. As the light passes through the sample, it acquires a complex amplitude that depends on the optical properties of the materials it traverses, generating inline holograms (*Figure 1—figure supplement 1*), which encode the three-dimensional position of the planktons as well as their size and refractive index. A CMOS camera located on the opposite side of the sample acquires the holograms for further analysis with a frame rate of $10\,\mathrm{fps}$, and an exposure time of $8\,\mathrm{ms}$.

In order to measure the position and dry mass of the planktons, the recorded holograms are analyzed by a regression U-Net (RU-Net, *Figure 1b* and *Figure 1—figure supplement 2*, see details in Methods, 'RU-Net architecture and training'). The RU-Net is a deep-learning architecture based on a modified U-Net, with two parallel arms in the upsampling path. The output of the RU-Net is a five-channel image where each channel corresponds to a heat map containing: a segmentation of the planktons from the background used to obtain a rough estimate of their $xy$ (in-plane) position; their estimated $z$ (axial) position; the plankton estimated dry mass $m$; and the distances $\Delta x$ and $\Delta y$ from the closest plankton for each pixel (used to improve the in-plane localization). This RU-Net is implemented and trained on simulated input–output image pairs (4000 samples) using the Python software package DeepTrack 2.0 (*Midtvedt et al., 2021a*). The output heat maps are finally processed to obtain a prediction of the plankton three-dimensional position and their dry mass, as shown in *Figure 1c*.

In order to increase the accuracy of the dry mass estimations, we extract time sequences of holographic images cropped around an individual plankton (*Figure 1d* and *Figure 1—figure supplement 3*) and further analyze them with a second deep-learning network. This is a weighted-average convolutional neural network (WAC-Net, *Midtvedt et al., 2021b*), *Figure 1e* and *Figure 1—figure supplement 3*, see details in Methods, 'WAC-Net architecture and training'. The WAC-Net determines a single estimated value of the equivalent spherical radius, as well as a more accurate value of the dry mass of the plankton in the sequence, through a weighted average of the latent representation of various holograms with learnable weights. The number of frames in the sequence is limited to 15 frames for training the WAC-Net. For inference, the length of the sequence is dependent on the application. For example, when analyzing feeding events we aim to capture dry mass dynamics on short time scales, and the sequence length is therefore restricted to a single frame. For the division

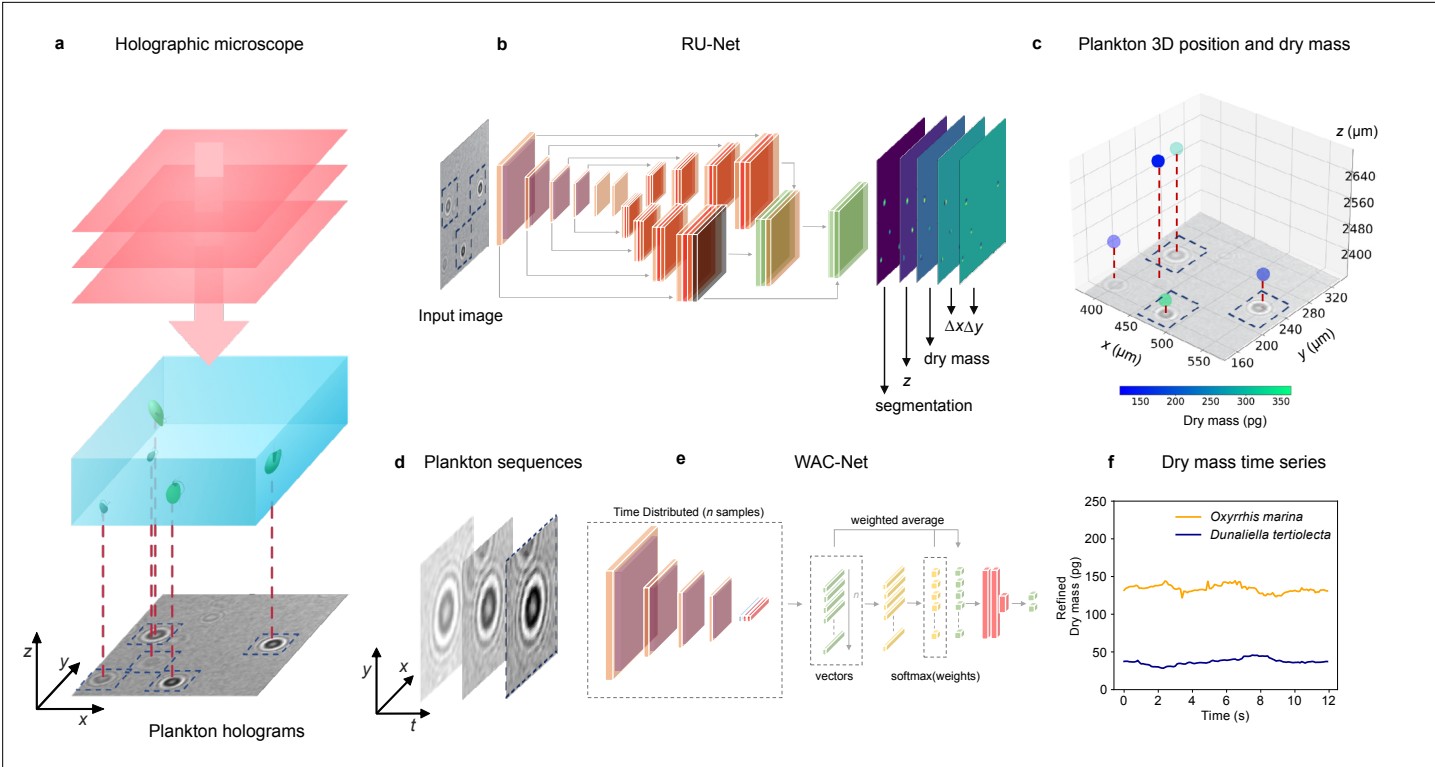

**Figure 1.** Experimental setup and deep-learning data analysis. (a) Holographic microscope: Planktons suspended in a miniature sample well are imaged with an inline holographic microscope. The (cropped) example holographic image features two different plankton species: *Oxyrrhis marina* and *Dunaliella tertiolecta* (full image in *Figure 1—figure supplement 1*). (b) Deep-learning network 1: A regression U-Net (RU-Net, see details in *Figure 1—figure supplement 2*), trained on simulated holograms, uses individual holograms to predict output maps containing the segmentation of the planktons, their $z$-position, their dry mass $m$, and the distances $\Delta x$ and $\Delta y$ from the closest plankton for each pixel (to be used for the accurate localization of planktons). (c) Plankton 3D position and dry mass: The information obtained by the RU-Net permits us to reconstruct the 3D position of the planktons along with their dry mass (color bar). (d) Plankton sequences: Using the plankton positions obtained by the RU-Net, we extract sequences of $64 \times 64$-pixel holograms centered on an individual plankton. (e) Deep-learning network 2: The sequences are then used by a weighted-average convolutional neural network (WAC-Net, see details in *Figure 1—figure supplement 3*), trained on simulated data, to refine the estimations of $m$ and $z$. (f) Dry mass time series: Example of a refined dry mass prediction in picograms (pg) for a micro-zooplankton (*Oxyrrhis marina*, orange line) and a phytoplankton (*Dunaliella tertiolecta*, blue line) obtained by the WAC-Net.

The online version of this article includes the following figure supplement(s) for figure 1:

**Figure supplement 1.** Holographic microscope and full-scale view of an experimental holographic image.

**Figure supplement 2.** Simulated holographic images and RU-Net architecture.

**Figure supplement 3.** Simulated plankton sequence and weighted-average convolutional neural network (WAC-Net) architecture.

events, the sequence length is 15 frames, as they occur over longer times ranging from hours to days with more recorded frames. Also the WAC-Net is implemented and trained with simulated data (4000 15-frame sequences of $64\,\text{px} \times 64\,\text{px}$ images) using DeepTrack 2.0 (*Midtvedt et al., 2021a*). *Figure 1f* shows an example of the dry mass output of the WAC-Net in picograms (pg) when applied on a sliding window over a sequence of holograms corresponding to a micro-zooplankton (*Oxyrrhis marina*) and a phytoplankton (*Dunaliella tertiolecta*).

## Dry mass estimates

The combination of RU-Net and WAC-Net permits us to measure the dry mass of each plankton at any point in time. For example, *Figure 2a* shows a portion of an inline hologram of the micro-zooplankton species, *O. marina*, tracked by the RU-Net (circles). Individual *O. marina* cells are then traced for 30 frames and their holograms are further processed with WAC-Net to obtain an estimation of the dry mass for each cell. The orange histogram in *Figure 2b* shows the dry mass distribution estimated by WAC-Net.

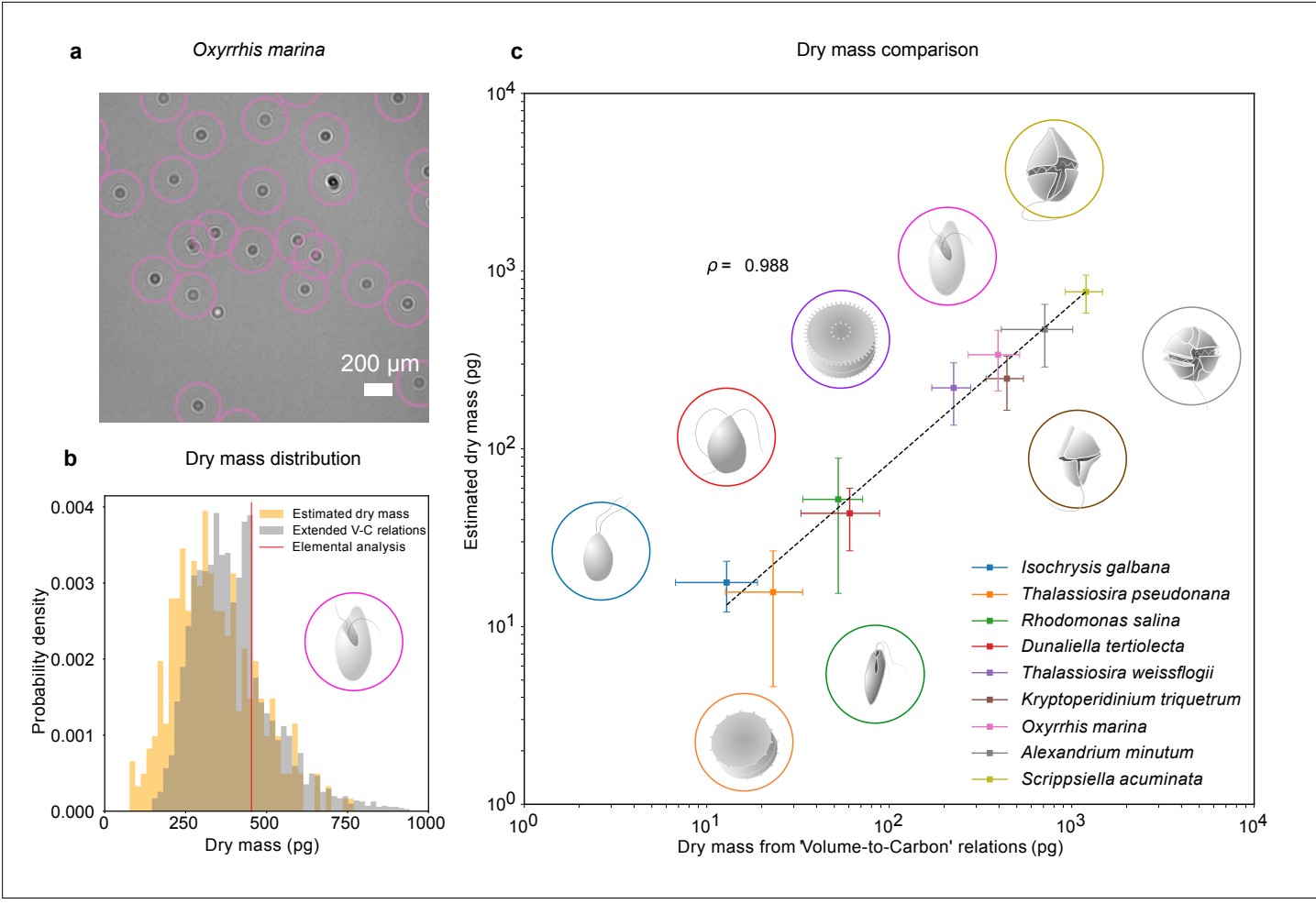

**Figure 2.** Dry mass estimates. (a) Phytoplankton species *Oxyrrhis marina* as detected by RU-Net on a portion of experimental hologram (see *Figure 1—figure supplement 1* for the complete hologram). (b) Dry mass distributions for *O. marina* (illustrated in the inset) obtained by applying weighted-average convolutional neural network (WAC-Net) to the experimental holograms (orange) and by volume-to-carbon relationships (gray, *Menden-Deuer and Lessard, 2000*). The red line is the value of the average mass estimate obtained from elemental analysis. (c) Comparison of the dry mass estimations obtained by WAC-Net and by the volume-to-carbon method for nine different species of diatoms (*Thalassiosira pseudonana, Thalassiosira weissflogii*), phytoplantons (*Isochrysis galbana, Rhodomonas salina, Dunaliella tertiolecta*), and micro-zooplanktons (*Oxyrrhis marina, Kryptoperidinium triquetrum, Alexandrium minutum, Scrippsiella acuminata*). The two measurements have a correlation coefficient of $\rho = 0.988$. The dashed line represents the best fit and the error bars show the standard deviations of the distributions. The insets illustrate each species.

The online version of this article includes the following figure supplement(s) for figure 2:

**Figure supplement 1.** Dry mass and equivalent spherical radius (ESR) estimates for different species of planktons.

To benchmark the dry mass measurements, we used the volume-to-carbon relationships from *Menden-Deuer and Lessard, 2000* followed by an extrapolation of elemental composition using extended Redfield ratios (*Anderson, 1995* see Methods, 'Dry mass estimation by volume-to-carbon relationships'). The gray histogram in *Figure 2b* shows the results for the case of *O. marina*. The dry mass predicted by the volume-to-carbon relationships (394 ± 123 pg, the uncertainty represents the standard deviations of the distribution) agrees well with the dry mass estimated by our technique (338 ± 126 pg, orange histogram). Importantly, in contrast to the volume-to-carbon relation method, the dry mass estimated by our approach can be tagged to individual cells in the image. This additional feature can be used to study the dry mass evolution of single cells (e.g., in the following sections, we will exploit this possibility in two exemplary studies of feeding and cell division events).

We repeated this analysis for nine species of planktons belonging to different taxonomic groups and trophic levels in the marine ecosystem (see Methods, 'Plankton cultures'): phytoplankton species (*Isochrysis galbana, Rhodomonas salina, Dunaliella tertiolecta*); micro-zooplankton species

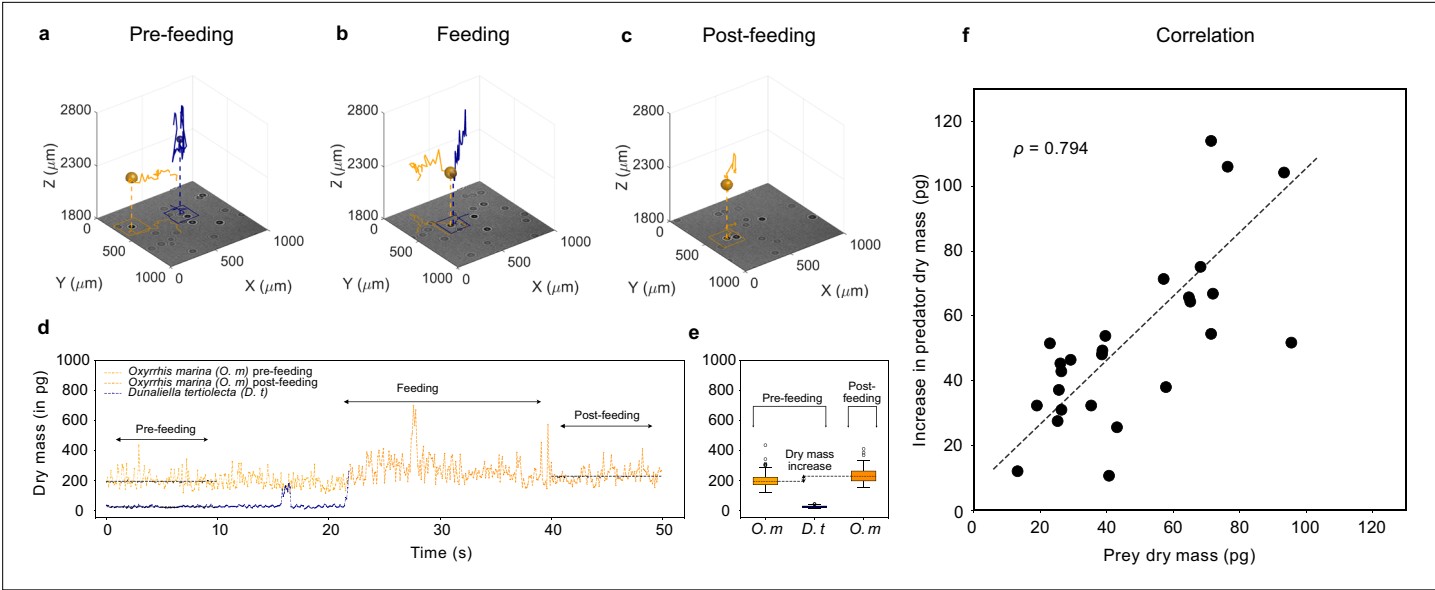

**Figure 3.** Feeding events. 3D recording of a feeding event where (a) a predator micro-zooplankton (*Oxyrrhis marina*, orange traces) approaches a prey phytoplankton (*Dunaliella tertiolecta*, blue traces), (b) feeds on it, and (c) finally moves away (see *Video 1* and *Figure 3—figure supplement 2*). The 2D projection of traces is superimposed on the holographic images in the bottom (see also *Figure 3—figure supplement 1*). (d) Dry mass time series of predator (orange trace) and prey (blue trace) estimated by weighted-average convolutional neural network (WAC-Net) in the three different phases. (e) The pre-feeding dry mass distributions of the predator *Oxyrrhis marina* (*O. m*) and the prey *Dunaliella tertiolecta* (*D. t*), and the post-feeding dry mass distribution of predator are represented in the box plots. The dry mass increase between pre- and post-feeding phases of the predator is indicated in the plot. The post-feeding dry mass increment of the predator (*O. m*) matches the dry mass of the prey (*D. t*). (f) There is a high correlation ($\rho = 0.794$) between dry mass increments of predators and dry mass of prey for 26 feeding events. The dashed line represents the best fit.

The online version of this article includes the following figure supplement(s) for figure 3:

**Figure supplement 1.** 2D projection of traces in a feeding event.

**Figure supplement 2.** Feeding event main.

**Figure supplement 3.** Feeding event additional.

(*Kryptoperidinium triquetrum*, *Alexandrium minutum*, *Scrippsiella acuminata*, along with *Oxyrrhis marina* which is used in the above discussion); and diatomic species (*Thalassiosira weissflogii*, *Thalassiosira pseudonana*). These results are summarized in *Figure 2c*. The data points and error bars represent the means and standard deviations of the dry mass distributions estimated by our method and the volume-to-carbon method. The two estimates correlate very well (correlation coefficient $\rho = 0.988$). A detailed dry mass distribution comparison (along with equivalent spherical radius distribution comparison) for different species can be seen in *Figure 2—figure supplement 1*.

As a further independent test, we also estimated the dry mass from the elemental analysis of carbon and nitrogen content in *O. marina* (extrapolated to the other fundamental elements hydrogen, oxygen, and phosphorous through Redfield ratios, *Anderson, 1995*, see Methods, 'Dry mass estimation by elemental analysis'). The resulting dry mass (453 pg, indicated with a red line in *Figure 2*) also confirms that our method arrives at realistic numbers. The average value indicated by the red line in *Figure 2b* lies within the distributions predicted by holographic estimate.

## Feeding events

We use the phytoplankton species *D. tertiolecta* and the micro-zooplankton species *O. marina* as the prey and predator, respectively. *Figure 3a–c* shows the 3D traces of prey (blue) and predator (orange) during a feeding event (see 3D movie of

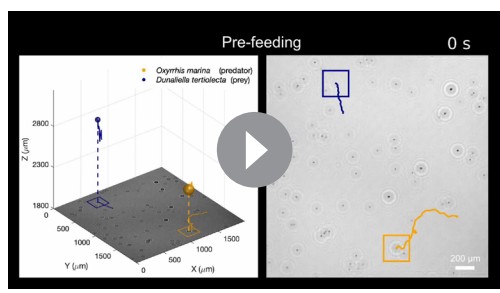

**Video 1.** Feeding event 1.
https://elifesciences.org/articles/79760/figures#video1

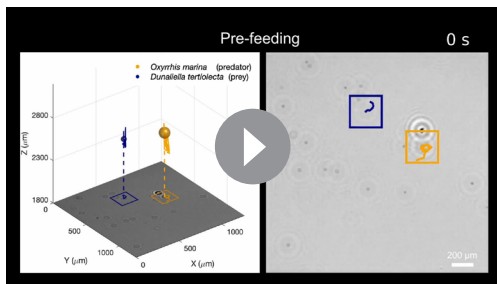

**Video 2.** Feeding event 2.
https://elifesciences.org/articles/79760/figures#video2

the feeding event in *Video 1*). In the pre-feeding phase *Figure 3a*, corresponding to about 10 s or 100 frames (see also *Figure 3d*), the predator explores the sample volume in a random fashion. It passes the prey cell closely on a couple of occasions before it makes contact (see *Videos 1 and 2* and , *Figure 3—figure supplement 2*, and *Figure 3—figure supplement 3*). In the feeding phase (*Figure 3b*, lasting for about 20 s or 200 frames), the predator makes contact with the prey and performs a localized swirling motion about a fixed location for 16 s while handling the prey. In the post-feeding phase (*Figure 3c*, last 10 s or 100 frames, see also *Figure 3d*), the predator returns back to its normal swimming behavior and carries on its search for new prey.

*Figure 3d* shows the dry mass time series of prey and predator during the feeding event. As the feeding events happen on a short time scale compared to the frame rate of the camera, we use WAC-Net with a sliding window of only one frame, maximizing the available temporal resolution of the dry mass estimation. The dry mass distributions of the prey and predator in pre- and post-feeding phases are shown by the box plots (*Figure 3e*) to the right hand side. In the pre-feeding phase: the prey dry mass is measured to be 26 ± 1 pg (blue box plot) and the predator 204 ± 5 pg (orange box plot). The uncertainties represent the standard error of the mean. The post-feeding dry mass distribution of the predator is 234 ± 5 pg. The difference in predator dry mass post- and pre-feeding closely matches the prey dry mass (*Figure 3e*). This indicates that the predator has fully consumed its prey, thus providing a direct measurement of the amount of the dry mass consumed during each individual feeding event.

In *Figure 3f*, we report the results of the dry mass increase in 26 feeding events. The increase in the predator dry mass in the post-feeding phase correlates well with the pre-feeding dry mass of the prey (correlation coefficient $\rho = 0.794$). The slope of the best fit line (with slope, $\alpha = 0.97$) also indicates that on average 97% of prey is consumed by the predator in a feeding event. Thus, it is possible to quantify individual feeding rates and, if predator cells are followed over time, also gross growth efficiency, that is, how much of the consumed biomass is converted into predator biomass.

## Life history of a plankton

The technique we have developed can follow the entire life histories of planktons, over time scales from hours to days. To demonstrate this, we use a diatom species, *T. weissflogii*, which is autotrophic and nonmotile. Over a preriod of 8 hr (*Figure 4*), we image a *T. weissflogii* and two generations of its daughter cells, continuously assessing the changes in their dry mass using the WAC-Net, which we already used to estimate the dry mass of *T. weissflogii* in *Figure 2c* (see Methods, 'Holographic imaging'). We place a low-density (1000 cells ml$^{-1}$) culture of diatoms in the sample well, which we illuminate with a white light source (5 W, 60 Hz warm light source bulb, aligned not to affect the holographic imaging sensor) to aid the cell growth.

*Figure 4a–e* shows the growth and division of a diatom imaged over a small portion of the sample. The parent cell (highlighted in *Figure 4a*) initially divides into two daughter cells, approximately 0.14 hr into the experiment (*Figure 4b*). Note that the biomass does not divide equally between the daughter cells. Asymmetric division in terms of cell size had already been shown in both bacteria and diatoms; our experiments now show that the daughter cells receive unequal proportions of the biomass from the mother cell. Then, the two daughter cells move slightly apart (*Figure 4c*) and the cell with the largest biomass of the two divides again at 4.86 hr (*Figure 4d, e*).

*Figure 4f* shows the dry mass of the parent and daughter cells as the experiment proceeds. We remark that, while the dry mass of these cells is continuously monitored, the WAC-Net estimates the most reliable values when the cells are isolated. Therefore, we consider the reference dry mass measurements as those when the cells have at least 3.6 μm (40 px) of empty space around them before or after each division; these times are indicated by the gray dashed lines in *Figure 4f*. The initial parent cell dry mass (measured at 1.1 hr) is estimated at 433 ± 2 pg. The dry mass of its two daughter cells

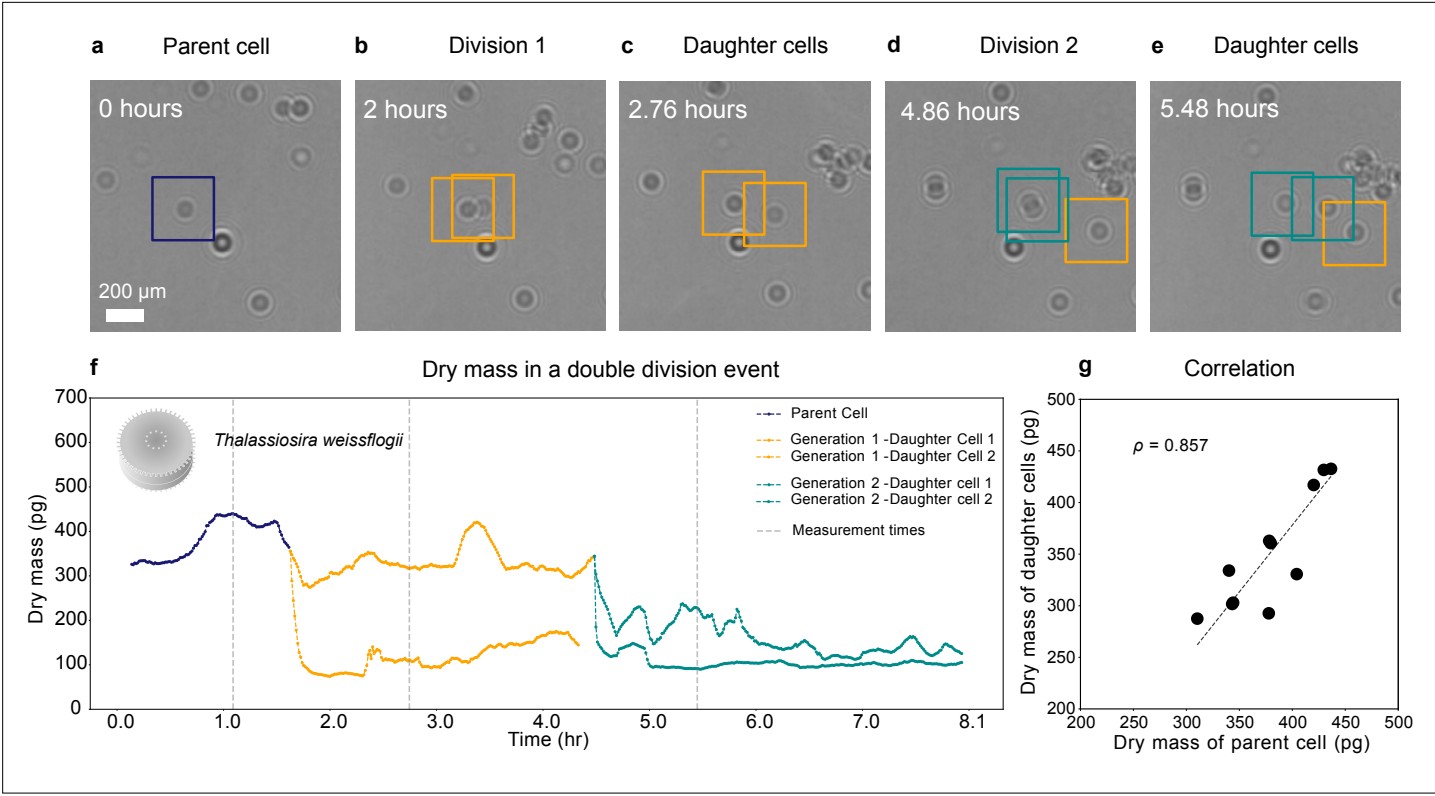

**Figure 4.** Growth and cell division of a diatom. Different life stages of a diatom (*Thallasiosira weissflogii*) and its daughter cells: (a) the parent cell (blue), (b) divides into two daughter cells (orange); (c) the daughter cells continue to grow, (d, e) until another cell division occurs (green). (f) Dry mass time series through generations estimated by weighted-average convolutional neural network (WAC-Net) (see also *Video 3* and *Figure 4—figure supplement 1*). Each cell dry mass is estimated when it has at least 3.6 µm (40 px) of empty space around it to ensure optimal performance of the WAC-Net; the corresponding times are indicated by the gray dashed lines. A drop in the dry mass values can be noticed with the daughter cells in subsequent divisions. (g) Correlation plot showing the relation between the sum of the dry masses of the daughter cells and the dry mass of the parent cell for 11 different division events ($\rho = 0.857$). The dashed line represents the best fit.

The online version of this article includes the following figure supplement(s) for figure 4:

**Figure supplement 1.** Division event.

(measured at 2.7 hr, as soon as the two daughter cells move sufficiently apart) is 326 ± 1 pg and 110 ± 1 pg, whose sum is close to the dry mass of the parent cell. As the experiment proceeds, one of the daughter cells divides again producing a second generation of daughter cells (*Figure 4d*), whose dry masses are 225 ± 3 pg and 93 ± 1 pg (at 5.48 hr, *Figure 4e*). Again, their sum is close to the mass of their parent cell. The uncertainty in the dry mass value represents the standard error of the mean computed for ±5 frames around the measurement point (gray dashed lines in *Figure 4*).

We have repeated this experiment with various cell densities with independently cultured samples, collecting multiple division events. *Figure 4g* shows the high correlation ($\rho = 0.857$) between the parent cell dry mass and the sum of the daughter cells' dry masses for 11 cell divisions. It is interesting to note that the division events of *T. weissflogii* occur when the parent cell weighs between 310 pg and 436 pg, with a mean value of ≈ 378 pg. This kind of a tip-off value prediction in dry mass for a division event is achieved for the first time thanks to this method and is another example of the type of information that

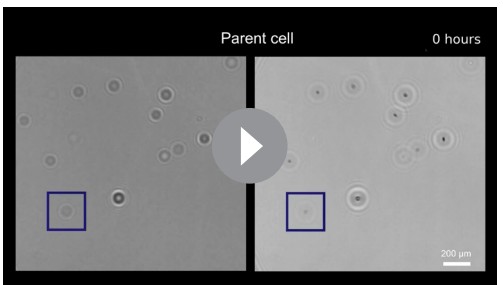

**Video 3.** Division event.
https://elifesciences.org/articles/79760/figures#video3

can be acquired by employing a nonintrusive technique that can continuously measure single cells throughout the cell cycle.

## Discussion

The main advantage of combining holographic microscopy with deep-learning algorithms lies in the ability to monitor position and dry mass of individual plankton cells over extended time periods. The method is nondestructive and minimally invasive, and allows quantitative assessment of trophic interactions such as feeding and biomass increase throughout the cell cycle, providing unprecedented detail to the life histories of marine microorganisms.

The standard methods to determine the biomass of cells entail either performing elemental analysis on cells harvested from single species cultures or estimating the biomass from volume-to-carbon relationships drawn from multiple elemental analyses of similar plankton organisms of different sizes (*Strathmann, 1967*; *Menden-Deuer and Lessard, 2000*). Elemental analysis has the advantage of providing detailed measurements of individual elements, typically carbon, nitrogen, and hydrogen; however, it is destructive and cannot provide individual cell resolution. Volume-to-carbon relationships can provide biomass estimates of individual living cells as long as the volume of the cells can be measured accurately (*Menden-Deuer and Lessard, 2000*); yet, the variability around the relationship is substantial (e.g., the estimated value for *O. marina* used in *Menden-Deuer and Lessard, 2000* is 30% higher than that measured by elemental analysis). Moreover, the volume-to-carbon relationships do not account for the nutritional status of the cell (e.g., as discussed in Results, 'Dry mass estimates', similarly sized cells of the same species can indeed differ by more than a factor of two in the estimated dry mass by the volume-to-carbon method).

*Figure 4f* reveals asymmetric cleavage in *T. weissflogii*, where sister cells receive unequal proportions of the mother cell biomass. Asymmetric division is well established in centric diatoms that have a rigid silica shell called the frustule consisting of two halves. Both halves have the shape of a cylinder with one open end; the smaller one (known as hypotheca) fits inside the larger one. Upon division, both daughter cells grow a new hypotheca fitting inside the half that is inherited from the mother cell so that one of the daughter cells will be slightly smaller (while the other maintains the size of the mother cell). When the cells reach a critically small size, the size is reinstalled through sexual reproduction (*Macdonald, 1869*; *Pfitzer, 1869*). Experimentally, asymmetric cleavage beyond the reductive cell cycle has also been shown in the diatom *Ditulum breightwelli* (*Laney et al., 2012*) where daughter cells are of different volume. Similar to our study, the sister cells show unequal times to the subsequent division (*Figure 4a–e*) which may lead to faster population growth (*Laney et al., 2012*). Physiological asymmetry is also found in bacteria (e.g., *E. coli*) where old damaged cell content is distributed differently leading to the development of age structure in prokaryote populations (*Proenca et al., 2018*). The differences in biomass of sister cells observed here are larger than expected from the volume alone which suggest that unequal division of biomass may be more common in protozoans than previously perceived.

Three-dimensional tracking of microorganisms is not easily achieved by alternative methods. In many instances, two-dimensional traces are obtained and three-dimensional swimming behavior is inferred by assuming isotropic swimming (*Selander et al., 2011*). Moreover, traditional tracking techniques often lose track of cells when they intersect other cells or swim out of focus. Holographic microscopy which records the holograms of objects overcomes the limitations due to the shallow depth of field in conventional light microscopy, and the three-dimensional positioning together with the biomass estimates facilitates linking of cell positions into coherent trajectories. The RU-Net approach (*Figure 1—figure supplement 2c*) that we describe can be a useful candidate to track inter-esecting cells from a recorded holographic image. Three-dimensional positioning of both prey and predator cells also allows detailed observations of cell–cell interactions such as reaction distances and rejection events. Finally, holographic microscopy does not require manipulation such as dilution and is not sensitive to the same assumptions as the dilution technique. On the other hand, the volume that can be monitored by holographic microscopy is limited by the coherence length of the light source. Here, we use an LED light source with a relatively short coherence length (approximately 200μm with 1-nm line filter). The depth of the observational chamber can, however, be increased by use of laser light sources with longer coherence length. With simple modifications to the setup design, the lens-less approach used here can be adopted for smaller organisms such as bacteria and heterotrophic

nanoflagellates, or larger organisms such as rotifers and small crustaceans. It could also be merged with rotating stages that keep cells in suspension (*Krishnamurthy et al., 2020*), or microfluidic channels to facilitate experiments with large number of cells in flow. The method is particularly suitable to study micro-zooplankton grazing behaviors.

A large and growing proportion of micro-phytoplanktons previously considered fully autotrophic have been reclassified as mixotrophic, that is, supporting their energy demand by a combination of photosynthesis and uptake of dissolved or particulate organic matter (*Stoecker et al., 2017*). This discovery is of more than academic interest as allowing mixotrophy in food web modeling results in up to a threefold increase in average organism size and enhanced transfer of biomass to higher trophic levels, thus increasing the sinking flux of carbon, 'the biological pump', by an estimated 35% (*Ward and Follows, 2016*). The discovery has led to something of a paradigm shift in marine microbial ecology and highlighted the need for new methods to accurately account for mixotrophy in biogeochemical models (*Flynn et al., 2019*). Mixotrophy is a plastic trait that changes with conditions. Among the more extreme cases, there is the dinoflagellate *Karlodinium armiger*, which is an autotroph at low cell concentrations, but switches to heterotrophy at high densities and even revert the food web by killing their copepod predators and extracting their content through peduncle feeding (*Berge et al., 2012*). The combination of holographic microscopy and deep-learning algorithms can be used to quantify uptake of both particulate (*Figure 3*) and dissolved matter. Uptake of dissolved organic matter will result in increased dry mass, and the slope of such increase will provide cell-specific uptake rates. Furthermore, it will allow us to explore the level of mixotrophy in different conditions and organisms by monitoring dry mass in factorial combinations of, for example, light and organic substrates. For instance, primary production can be assessed as dry mass increase in the presence of light but the absence of organic food sources (*Figure 4*). In the scenarios where both photosynthesis and uptake of organic food sources are possible, the level of heterotrophy can be estimated by subtracting photosynthesis from total production.

Individual-level observations have been key to further the development of ecological theory, not only in larger organisms, but also in smaller organisms such as millimeter-sized copepods, the most common multicellular organisms in the ocean. As an example, copepods have been shown to feed selectively and reject well-defended cells. Consequently, well-defended cells are favored and enriched by copepod grazing, which contributes to harmful algal bloom formation (*Ryderheim et al., 2021*). Moreover, individual-level observations have revealed the sensory capabilities of copepods involved in prey and threat detection as well as the fundamental strategies involved in foraging and reproduction (*Kiørboe et al., 1999*; *Kiørboe et al., 2009*). Likewise, Individual-level observations of protozoans have the potential to catalyze experiments and gain insights in microbial food–web interactions in a similar way.

We conclude that the marriage between holographic microscopy and deep learning provides a strong complementary tool in microbial ecology. It allows the nondestructive and minimally invasive determination of the three-dimensional position and dry mass of individual microorganisms. It outperforms traditional methods in terms of speed and individual resolution and rivals the precision and accuracy of current methods. While holographic microscopy has already been employed in marine sciences, the combination with deep-learning algorithms makes it more versatile and many orders of magnitude faster, which is key to follow and characterize individuals throughout their lifespan.

## Methods
### Holographic imaging
During the measurements, the planktons are placed under the holographic microscope (*Figure 1a*) in two different configurations depending on the application. For the short-time-scale experiments, such as the dry mass estimates in *Figure 2* and the feeding-event experiments in *Figure 3*, a sample of volume ≈40 µl is directly placed on a glass slide without any sample well to enclose the planktons. For the long-time-scale experiments, such as the division-event experiments in *Figure 4*, the diatoms are kept in enclosed circular wells (diameter 3 mm, depth 1 mm, volume ≈10 µl). As shown in *Figure 1—figure supplement 1a*, the planktons are imaged using a lensless holographic imaging technique (*Daloglu et al., 2018*), where the sample is illuminated by a narrow-band LED light source (Thorlabs M625L3, center wavelength 632 nm, bandwidth 18 nm with a 1-nm bandwidth line filter,

Thorlabs FL632.8-1, centered at $632.8\,\mathrm{nm}$) and the sensor (Thorlabs DCC1645C, CMOS sensor area $4.608\,\mathrm{mm} \times 3.686\,\mathrm{mm}$, $1024\,\mathrm{px} \times 1280\,\mathrm{px}$) is placed immediately below the bottom of the well (the distance from the bottom of the well to the sensor is $\approx 1.5\,\mathrm{mm}$). In this way, the entire well is imaged within a single field of view (*Figure 1—figure supplement 1b*), ensuring that all planktons are continuously visible for the whole duration of the experiment. The resulting images of the planktons are diffraction patterns formed by the interference of the unscattered light and the light scattered by the planktons. These diffraction patterns (holograms) act as a unique fingerprint of the size, refractive index, dry mass, as well as the lateral and axial position of the planktons, which has been used previously to characterize micron-scale objects (*Altman and Grier, 2020*). Physically, the dry mass can be defined as the difference between the mass of the object (here, the plankton) and the mass of an equal volume of medium (here, the watery solution). Thus, writing the mass concentration of biomolecules inside the object as $c_{\mathrm{object}}$, and the medium as $c_{\mathrm{med}}$, the dry mass of an object is given by $m_{\mathrm{dry}} = V(c_{\mathrm{object}} - c_{\mathrm{med}})$, where $V$ is the volume of the object. On the other hand, the light scattering at small angles is also proportional to the product $V(c_{\mathrm{object}} - c_{\mathrm{med}})$, making it possible to extract the dry mass of the objects from their holograms (see *Equation 5*, Appendix 1, 'Relation between dry mass and scattering cross-section').

## RU-Net architecture and training

Prior to the analysis by RU-Net and WAC-Net (explained in the next section), the diffraction patterns are normalized with respect to the intensity of the unscattered light. To obtain the plankton positions, we use a modified U-Net (*Ronneberger et al., 2015*), which we name Regression U-Net (RU-Net) implemented using DeepTrack 2.0 (*Midtvedt et al., 2021a*). Its architecture is shown in *Figure 1—figure supplement 2c*. The downsampling part of the RU-Net consists of a series of convolutional blocks, where each convolutional block contains a series of convolutional layers followed by a max-pooling layer and an ReLU activation. For an input image of size $128\,\mathrm{px} \times 128\,\mathrm{px}$ (about one-tenths the size of the acquired experimental image, *Figure 1—figure supplement 2a*), we use a sequence of six convolutional blocks containing 8, 16, 32, 64, 32, and 32 convolutional layers, respectively. In the upsampling part, the RU-Net is divided into two different paths that function as two independent regular U-Nets. Each upsampling path contains a series of four upsampling blocks with each containing a deconvolutional layer followed by a series of 128, 64, 32, and 16 convolutional layers. Features obtained from each convolutional block in the downsampling path are appended to the features of the upsampling path at each upsampling block. One of the upsampling paths is used for the segmentation of planktons for which a sigmoid activation is applied on the output of the final upsampling block. The other upsampling path is used to obtain the heat maps of dry mass, axial $z$-distance, and lateral $x$- and $y$-positions (to refine the lateral localization accuracy) for which a ReLU activation is applied. Finally, the outputs of both paths are concatenated to obtain a five-channel output tensor of size $128 \times 128 \times 5$ (*Figure 1—figure supplement 2b*).

To train the RU-Net, we simulate holographic images of size $128\,\mathrm{px} \times 128\,\mathrm{px}$ using DeepTrack 2.0 (*Midtvedt et al., 2021a*). Each image contains planktons of different sizes and refractive indices (*Figure 1—figure supplement 2a*). Planktons are simulated in a wide dry mass range from 1 pg to 995 pg, with their corresponding equivalent spherical diameters ranging from 1.5 μm to 10 μm. The RU-Net is trained using the AMSgrad optimizer (*Reddi et al., 2019*), with a learning rate of 0.0001. The model is trained on 4000 simulated holographic images in mini-batches of 16 images for 300 epochs, with a custom loss function. The images are generated with the continuous generator of DeepTrack 2.0 (*Midtvedt et al., 2021a*) starting with 2000 images generated before the training and the remaining images generated as the training proceeds. The training process (including the data generation) takes about 1.5 hr on a Kaggle server (Tesla P100 graphics processor unit and Intel(R) Xeon(R) CPU @ 2.00 GHz).

## WAC-Net architecture and training

To obtain a refined dry mass value, we use a WAC-Net (*Midtvedt et al., 2021b*) implemented using DeepTrack 2.0 (*Midtvedt et al., 2021a*). Its architecture is shown in *Figure 1—figure supplement 3b*. The downsampling part of WAC-Net contains a time-distributed block that consists of a series of convolutional blocks. Each convolutional block contains a convolutional layer followed by a ReLU activation and a max-pooling layer. For an input image sequence consisting of frames of size $64\,\mathrm{px} \times 64\,\mathrm{px}$,

we use a series of four convolutional blocks containing 32, 64, 128, and 256 convolutional filters, respectively. The features are then flattened and analyzed by a series of 2 dense layers with 128 nodes each to obtain the latent representations for the images in the sequence. We use two convolutional layers with 128 and 1 filters, respectively, on each of the output latent vectors to obtain single-value representations of the weights. These weights are further normalized with a softmax layer. We average the latent vectors with the normalized weights to obtain a weighted representation of the latent vectors. Finally, we use a series of dense layers with 32, 32, and 2 nodes on the output representations to generate the output values of dry mass and radius. The dry mass predicted by the WAC-Net is converted to natural mass units by using a specific refractive index increment value, $\frac{dn}{dc} = 0.21 \, \mathrm{ml \, g^{-1}}$ accounting for the average planktonic solute composition (*Aas, 1996*) (See Appendix 1, 'Relation between dry mass and scattering cross-section').

To train the WAC-Net, we simulate 15-frame sequences of $64 \, \mathrm{px} \times 64 \, \mathrm{px}$ images that contain a main plankton (whose dry mass and radius the WAC-Net will estimate, *Figure 1—figure supplement 3a*) near the center of the image, which randomly moves by ±3.6 µm (± 1 px) in the $xy$-plane and ±100 µm in the $z$-direction (since the frame is laterally centered on the plankton, the $xy$-plane movement is smaller than the $z$-movement). In order to make the network robust to the existence of multiple planktons within a frame, other planktons are occasionally added to the frames. These additional planktons are given a directed in-plane motion randomly chosen between 3.6 µm (1 px) and 25.2 µm (7 px) per frame. In the $z$-direction, the motion is also randomized at ±100 µm per frame. The WAC-Net is trained using the AMSgrad optimizer (*Reddi et al., 2019*), with a learning rate of 0.0001. The model is trained on 4000 images in mini-batches of 32 images for 200 epochs, with a mean absolute error (MAE) loss function. The images are generated with the continuous generator of DeepTrack 2.0 (*Midtvedt et al., 2021a*), starting with 2000 images generated before the training and with remaining images being generated as the training proceeds. The training process (including the data generation) takes about 45 min on a Kaggle server (Tesla P100 graphics processor unit and Intel(R) Xeon(R) CPU @ 2.00 GHz).

## Plankton cultures

We used a representative subset of plankton organisms covering larger primary producers such as diatoms (*T. weissflogii*, *T. pseudonana*) and dinoflagellates (*A. minutum*, *K. triquetrum*, *S. acuminata*) as well as smaller flagellates (*I. galbana*, *D. tertiolecta*). We also included the heterotrophic dinoflagellate *O. marina* to explore predator–prey interactions and feeding events (see *Table 1*, 'Planktons used in the experiments'). Plankton cultures were reared in L medium at 26 PSU salinity in a light and temperature controlled incubator (16 °C, 12 hr:12 hr light:dark cycles, $100 \, \mathrm{fmol \, m^2 s^{-1}}$). The *O. marina*

**Table 1.** Planktons used in the experiments.

Strain identifier denotes strain code in Gothenburg University Marine Algae Culture Collection (GUMACC) and synonym strain identifier in parenthesis. The *Oxyrrhis marina* culture was kindly provided by Denmark Technical University (DTU-Aqua) and does not have a strain ID. Equivalent Spherical Diameter (ESD) denotes the spherical diameter based on Coulter counts (Beckman multisizer III) of pure cultures.

| Scientific name | Strain identifier | Class | ESD (mean ± SD) |
|---|---|---|---|
| *Alexandrium minutum* | GUMACC83 (CCMP113) | Dinophyceae | 18.3 ± 2.5 µm |
| *Dunaliella tertiolecta* | GUMACC5 | Cholorphceae | 6.7 ± 0.9 µm |
| *Isochrysis galbana* | GUMACC108 (CCMP1323) | Prymnesiophyceae | 4.0 ± 0.7 µm |
| *Kryptoperidinium triquetrum* | GUMACC71 (LAC20, KA86) | Dinophyceae | 14.9 ± 1.2 µm |
| *Oxyrrhis marina* | DTU-Aqua | Dinophyceae | 14.5 ± 1.5 µm |
| *Rhodomonas salina* | GUMACC126 (CCAP978/27) | Cryptophyceae | 7.4 ± 1.0 µm |
| *Scripsiella acuminata* | GUMACC110 (CCMP1331) | Dinophyceae | 17.7 ± 4.9 µm |
| *Thallassiosira pseudonana* | GUMACC132 (CCAP1085/12) | Cosconodiscophyceae | 4.8 ± 0.9 µm |
| *Thallassiosira (Conticribra) weissflogii* | GUMACC162 (CCAP1085/18) | Cosconodiscophyceae | 12.9 ± 1.4 µm |

cultures were fed with *I. galbana* or *D. tertiolecta* weekly, but starved until prey cells became rare before experiments to avoid unintentional addition of prey cells to experiments.

## Dry mass estimation by volume-to-carbon relationships

We compare the dry mass estimates from the holographic microscopy against the standard method based on volume-to-carbon relationships by measuring the volume of the cells on a Coulter counter (Beckaman, Multisizer III). The Coulter counter is equipped with a 100 μm orifice tube. Its accuracy is confirmed with latex beads. Volume estimates are subsequently used to estimate the carbon content of the cells using the equations given in *Menden-Deuer and Lessard, 2000*.

## Dry mass estimation by elemental analysis

A precise algal culture volume of known cell concentration is filtered onto pre-combusted (450°C) 25 mm glass fiber filters (Whatman GF/F). The filters are dried overnight at 60°C. Carbonates are removed by incubation in an exicator together with fuming hydrochloric acid. The filters are enclosed in tin capsules and analysed on an elemental analyzer (ANCASL, SerCon, UK) coupled to an isotope ratio mass spectrometer (20–20, SerCon, UK).

## Data and code availability

All the relevant source code and the data are made publicly available at the Quantitative-Microplankton-Tracker repository (*Bachimanchi et al., 2022*).

## Acknowledgements

The authors would like to thank Jan Heuschele for the illustrations, and Olga Kourtchenko for providing plankton cultures. This work was partly supported by the H2020 European Research Council (ERC) Starting Grant ComplexSwimmers (Grant No. 677511), the Horizon Europe ERC Consolidator Grant MAPEI (Grant No. 101001267), the Knut and Alice Wallenberg Foundation (Grant No. 2019.0079), and the Swedish Research Council (VR, Grant No. 2019-05238)

## Additional information

### Funding

| Funder | Grant reference number | Author |
| --- | --- | --- |
| European Research Council | 677511 | Harshith Bachimanchi Benjamin Midtvedt Giovanni Volpe |
| European Research Council | 101001267 | Harshith Bachimanchi Benjamin Midtvedt Giovanni Volpe |
| Knut och Alice Wallenbergs Stiftelse | 2019.0079 | Harshith Bachimanchi Benjamin Midtvedt Giovanni Volpe |
| Vetenskapsrådet | 2019-05238 | Erik Selander |

The funders had no role in study design, data collection, and interpretation, or the decision to submit the work for publication.

### Author contributions

Harshith Bachimanchi, Conceptualization, Data curation, Software, Formal analysis, Investigation, Visualization, Methodology, Writing - original draft, Writing - review and editing; Benjamin Midtvedt, Software, Investigation, Methodology, Writing - review and editing; Daniel Midtvedt, Conceptualization, Software, Formal analysis, Supervision, Investigation, Methodology, Writing - review and editing; Erik Selander, Conceptualization, Resources, Data curation, Supervision, Funding acquisition, Investigation, Project administration, Writing - review and editing; Giovanni Volpe, Conceptualization,

Resources, Supervision, Funding acquisition, Methodology, Project administration, Writing - review and editing

### Author ORCIDs
Harshith Bachimanchi (ID) http://orcid.org/0000-0001-9497-8410
Giovanni Volpe (ID) http://orcid.org/0000-0001-5057-1846

### Decision letter and Author response
Decision letter https://doi.org/10.7554/eLife.79760.sa1
Author response https://doi.org/10.7554/eLife.79760.sa2

## Additional files

### Supplementary files
• MDAR checklist

### Data availability
All the relevant source code and data are made publicly available at Quantitative-Microplankton-Tracker GitHub repository (https://github.com/softmatterlab/Quantitative-Microplankton-Tracker, (copy archived at swh:1:rev:55bcfa872a9eca7145a65e98467f37ceffe31052)).

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

## Appendix 1

### Relation between dry mass and scattering cross-section

The amount of light scattered in any given direction from an object is a function of the shape and the refractive index of the object. The scattering amplitude $S(\theta)$ quantifies the amount of light that is deflected an angle $\theta$ from the direction of the illuminating light. For objects significantly larger than the illuminating wavelength ($2\pi R/\lambda \gg 1$, where $R$ is the radius of the object and $\lambda$ is the wavelength of the light) and small refractive index difference ($|n_{object}/n_{med} - 1| \ll 1$, where $n_{object}$ is the refractive index of the object and $n_{med}$ is that of the medium), the scattering amplitude in the far field region can be estimated within the anomalous diffraction approximation as (*Streekstra et al., 1993*),

$$S(\beta, \gamma) = \left(k^2/2\pi\right) \int \int \left[1 - \exp(-i\phi(\epsilon, \nu))\right] \times \exp(ik(\epsilon\beta + \nu\gamma))d\epsilon d\nu, \tag{1}$$

where $k$ is the light wave number, $\epsilon, \nu$ are the in-plane spatial coordinates at the axial plane of the object (the plankton), while $\beta, \gamma$ are the coordinates at the axial plane of the sensor, scaled by the distance $r$ between plankton and sensor position, $\beta = x/r$ and $\gamma = y/r$ with $r = (x^2 + y^2 + z^2)^{1/2}$. Parametrizing the scattering coordinates in spherical coordinates, one has that $\beta = \sin\theta\cos\varphi$, $\gamma = \sin\theta\sin\varphi$. Averaging the scattering amplitude over the angle $\varphi$ one obtains,

$$S(\theta) = k^2 \int \int \left[1 - \exp(-i\phi(\rho, \chi))\right] \times J_0(k\rho\sin\theta)\rho d\rho d\chi, \tag{2}$$

with $\rho = \sqrt{\epsilon^2 + \nu^2}$ and $\sin\chi = \epsilon/\rho$, while $J_0$ is the bessel function of the first kind of order 0. Since the planktons in our setup are measured in transmission, the signal relates primarily to small angle scattering. To lowest order in the scattering angle one obtains,

$$S(\theta) \approx S(0) \approx k^2 A \left(1 - \cos\langle\phi\rangle - i\sin\langle\phi\rangle\right), \tag{3}$$

where the spatial distribution of the phase shift over the particle is replaced by its average value, denoted by $\langle\cdot\rangle$, and $A$ is the cross-sectional area of the object. The average value of the phase shift can be can be directly related to the dry mass of the particle by noting that the local phase shift, $\phi(\epsilon, \nu)$, due to a weakly scattering object with refractive index $n$ is given by,

$$\phi(\epsilon, \nu) = Z(\epsilon, \nu)(n - n_{med}), \tag{4}$$

where $Z(\epsilon, \nu)$ is the local thickness of the object and $n_{med}$ is the refractive index of the solution.

Dry mass can also be defined as the difference between the mass of the object and the mass of an equal volume of the medium. Thus, writing the mass concentration of biomolecules inside the object as $c_{object}$ and that inside the medium as $c_{med}$, the dry mass of an object is given by $m_{dry} = V(c_{object} - c_{med})$, where $V$ is the volume of the object. Empirically, the mass concentration of a solution of biomolecules and the refractive index of the solute are linearly related as $c = (\frac{dn}{dc})^{-1}(n - n_{med})$, where $n$ is the refractive index of the solution, and $n_{med}$ is the refractive index of the medium in the absence of biomolecules (*Zangle and Teitell, 2014*).

As a consequence, one has that.

$$\langle\phi\rangle = \langle kZ\rangle \cdot (n - n_{med}) = kV/A \cdot c\frac{dn}{dc} = km/A \cdot \frac{dn}{dc}, \tag{5}$$

where $m$ is the mass of the biomolecules within the object, $\langle Z\rangle$ is the average thickness of the object, $V$ is the volume of the object, and $A$ is the cross-sectional area of the object. Thus, the scattering amplitude depends explicitly on the mass of biological objects.

In order to relate the scattering amplitude to the measured intensity of the scattering pattern, we note that the intensity is given by,

$$I = |E_0|^2 \left|1 - i\frac{S}{kr}\exp(ik(r-z))\right|, \tag{6}$$

where $E_0$ is the amplitude of the incoming light.

From this expression, the change in light intensity in the middle of the scattering pattern compared to the unscattered light, normalized by the intensity of the incoming light, is given by,

$$\Delta I(0) = \frac{I}{|E_0|^2} - 1 = (2/kz) \, \Im S = (2kA/z) \sin\left[km/A \left(dn/dc\right)\right].  \tag{7}$$

To lowest order in the phase shift $\phi$, the scattering intensity at 0 degree scattering angle is therefore directly proportional to the mass of a biological object.

This formalism also provides a direct means to assess the influence of absorption on the estimated mass. Light absorption is quantified by the imaginary part of the refractive index. Adding an imaginary part to the refractive index in **Equation 5**, such that $n = n_0 + i\eta$, the light intensity in the forward direction becomes,

$$\Delta I(0) = (2kA/z) \sin\left[km/A \left(dn/dc\right)\right] \exp\left(-k\eta\langle Z \rangle\right).  \tag{8}$$

Using $\eta = 0.002$ as a typical value for phytoplankton (**Qi et al., 2016**), and an average thickness of 5 µm, the correction due to absorption would amount to about 10%.

