## [Editor Report]

This paper presents a valuable new method combining holographic microscopy and deep learning to track the behavior and growth of individual plankton. The paper illustrates the method with compelling data from two applications, zooplankton feeding behavior and diatom cell division. This paper will be of interest to plankton ecologists and ocean ecosystem modelers. The results obtained from this method will provide new insights into the trophic strategies of ocean plankton and important constraints for global ocean models.

---

## [Decision Letter]

**Decision letter after peer review:**

Thank you for submitting your article "Microplankton life histories revealed by holographic microscopy and deep learning" for consideration by *eLife*. Your article has been reviewed by 2 peer reviewers, one of whom is a member of our Board of Reviewing Editors, and the evaluation has been overseen by Meredith Schuman as the Senior Editor. The following individual involved in the review of your submission has agreed to reveal their identity: Fernando Peruani (Reviewer #2).

Essential revisions:

Both reviewers think the work is interesting and high-quality. The main issue raised by both is the ecological relevance of the work. A revised manuscript must address the following:

1) How restricted is the method to idealized conditions (e.g. cultured isolates, cell concentrations, etc)? What are the constraints in applying this method to more ecologically relevant conditions?

2) How can the results obtained by this method inform larger macroscale questions in plankton ecology?

*Reviewer #1 (Recommendations for the authors):*

In this manuscript, the authors sought to develop and apply a new method for tracking individual plankton over time. The method combines holographic microscopy and deep learning to produce time series of plankton positions in 3D space and dry mass. The authors demonstrate the power of their method in two case studies: feeding events in a two-species culture and cell division events in a diatom culture. The results from the first case study show feeding behavior and the increase in cell mass upon prey consumption, while the cell division study shows asymmetric division and a second round of division by the larger daughter cell. The authors suggest this method could be useful for studying growth efficiency, cell physiology, selective grazing, and mixotrophy.

The data presented are compelling and interesting, and clearly demonstrate the utility and power of the method. As a marine microbiologist, I cannot comment on the deep learning methods, though I found the description of the methods to be fairly clear.

This method seems to be extremely powerful and elegant, and the time series data are beautiful. It is amazing to see such detailed data for single-cell behaviours and interactions. I believe this method will be very useful for studying the biology of marine protists and micro/nanozooplankton, which are vastly underexplored. It will also be useful for studying microbial food webs and trophic transfer. Whether this method can be extended from cultures to field samples remains to be seen, but if possible would allow the study of uncultivated organisms in diverse habitats.

*Reviewer #2 (Recommendations for the authors):*

The authors develop a method, that combines holographic microscopy with deep learning, to monitor the dynamics of plankton cells for extended time periods (e.g. 8h). This is an important step to getting a better quantitative understanding of plankton systems. On the other hand, the method seems to allow only monitoring the plankton activity in reduced volumes and making use of a relatively slow sampling rate. The tracking of individual cells, possible via deep learning, seems that it cannot provide details on the predator-prey dynamics occurring at shorter time scales. Also, it seems that the method applies exclusively to very diluted systems. These limitations suggest that though the method can provide unique information on plankton systems, the ecological significance of such results is restricted to probably too idealized scenarios.

In the abstract and introduction, the authors motivate their study from a too macroscopic perspective of the problem. However, the limitation of the method seems to restrict the application of the method to idealized scenarios, whose ecological relevance is questionable. However, I believe that studying such idealized, small systems is an initial key step. I think the authors could try to discuss in more detail how such studies can help to understand the ecology of plankton systems.

---

## [Author Response]

Essential revisions:Both reviewers think the work is interesting and high-quality. The main issue raised by both is the ecological relevance of the work. A revised manuscript must address the following:

We thank the editor for the positive feedback on the article. We have responded to reviewers’ comments in detail in the reviewer specific sections of the response letter. We summarize the responses here for convenience.

1) How restricted is the method to idealized conditions (e.g. cultured isolates, cell concentrations, etc)? What are the constraints in applying this method to more ecologically relevant conditions?

There are certain limitations to the method that arise from the optics (See response to R1.2 and the revised manuscript with highlighted changes). For instance, the amount of sample volume that can be imaged is limited by the optics of the holographic microscope. We have indicated different ways to increase the volume in the “Discussion” section, for instance by using a source with a longer temporal coherence length.

Field samples are inherently more complex than the cultured isolates, with many more types of cells of overlapping size and dry mass. We have tested the technique with mixed samples containing two different plankton species (see feeding-event experiments in Figure. 3). The RU-Net is useful in this case to separate cells from other matter, apart from classifying the species. The continuous measurements of size / dry mass and three-dimensional position will allow us to quantitatively follow individual cells belonging to a certain class also in more complex environments. But it is true that there is an upper limit for the number of cells that can be imaged. We have tested the limits of the system, for instance in the feeding-event experiments (Figure. 3) where we have used a high concentration of prey cells (~700 cells observed simultaneously). It is observed that beyond a certain threshold, the accuracy in dry mass measurements deteriorates significantly due to multiple overlapping plankton holograms.

We also discuss some other possible limitations of the method in response to reviewer’s comment R1.2, where we also indicate the possible ways to overcome these limitations.

2) How can the results obtained by this method inform larger macroscale questions in plankton ecology?

We have added some specific examples in the “Discussion” section, where individual-level observations have furthered the understanding for higher-level dynamics. We mention how feeding preferences can structure marine communities, and exemplify with the insights gained from individual level observations of the larger mesozooplankton grazers, such as copepods.

Copepods have been shown to feed selectively and reject well-defended cells. Consequently, well-defended cells are favored and enriched by copepod grazing, which contributes to harmful algal bloom formation. Moreover, individual-level observations have revealed the sensory capabilities of copepods involved in prey and threat detection as well as the fundamental strategies involved in foraging and reproduction. Individual-level observations of protozoans have the potential to catalyze experiments and gain insights in microbial food-web interactions in a similar way (see response to R1.1 and R2.1. and revised manuscript file with highlighted changes).

Reviewer #1 (Recommendations for the authors):In this manuscript, the authors sought to develop and apply a new method for tracking individual plankton over time. The method combines holographic microscopy and deep learning to produce time series of plankton positions in 3D space and dry mass. The authors demonstrate the power of their method in two case studies: feeding events in a two-species culture and cell division events in a diatom culture. The results from the first case study show feeding behavior and the increase in cell mass upon prey consumption, while the cell division study shows asymmetric division and a second round of division by the larger daughter cell. The authors suggest this method could be useful for studying growth efficiency, cell physiology, selective grazing, and mixotrophy.The data presented are compelling and interesting, and clearly demonstrate the utility and power of the method. As a marine microbiologist, I cannot comment on the deep learning methods, though I found the description of the methods to be fairly clear.This method seems to be extremely powerful and elegant, and the time series data are beautiful. It is amazing to see such detailed data for single-cell behaviours and interactions. I believe this method will be very useful for studying the biology of marine protists and micro/nanozooplankton, which are vastly underexplored. It will also be useful for studying microbial food webs and trophic transfer. Whether this method can be extended from cultures to field samples remains to be seen, but if possible would allow the study of uncultivated organisms in diverse habitats.

We thank the reviewer for the positive assessment of our work and for the useful comments which helped us to improve the manuscript.

We would also like to highlight that all the code that has been used in this work is stored in our GitHub repository (https://github.com/softmatterlab/Quantitative-Microplankton-Tracker). There, we provide “training-tutorials”, where we give detailed examples on how to train the neural networks that are used in this work, and we provide “examples”, where we demonstrate how to use the trained neural networks to analyze the holographic data and subsequently generate the figures featured in the manuscript.

Reviewer #2 (Recommendations for the authors):The authors develop a method, that combines holographic microscopy with deep learning, to monitor the dynamics of plankton cells for extended time periods (e.g. 8h). This is an important step to getting a better quantitative understanding of plankton systems.

We thank the reviewer for the positive assessment of our work.

On the other hand, the method seems to allow only monitoring the plankton activity in reduced volumes and making use of a relatively slow sampling rate.

We agree that there are certain limitations to the method. As rightly pointed out by the reviewer, the amount of sample volume that can be imaged is limited by the optics of the holographic microscope (explained in more detail in response to comment R1.2). We have indicated different ways to increase the volume in the “Discussion” section, for instance by using a source with a longer temporal coherence length. And the throughput can be increased by using microfluidic channels by observing planktons in a flow, e.g., to get better dry-mass statistics.

The tracking of individual cells, possible via deep learning, seems that it cannot provide details on the predator-prey dynamics occurring at shorter time scales.

The temporal resolution is only limited by the frame rate the holograms are obtained, here typically around 10 holograms per second, so the method is not limited in this sense. The choice of this frame rate is a compromise between temporal resolution, experiment length, and quantity of image data to be stored. The holographic data can be easily acquired at a higher frame rate to improve the temporal resolution. The analysis via deep learning will still be readily possible without any modifications to the analysis pipeline.

Also, it seems that the method applies exclusively to very diluted systems. These limitations suggest that though the method can provide unique information on plankton systems, the ecological significance of such results is restricted to probably too idealized scenarios.

Observations in denser and/or more complex natural samples is indeed a challenge. Given the ability to keep track of the dry mass and 3D positions, we may still be able to follow individual cells in more complex settings. However, even in simpler laboratory settings individual-level resolution of trophic interactions have provided important insights for slightly larger organisms such as millimeter-sized copepods (see response to R1.1 and R1.2. and revised manuscript file with highlighted changes). These include detailed characterization of, e.g., feeding strategies, feeding preferences, handling times, sensory capabilities determining reaction distances, encounter rates in copepods. We hope to have now more clearly addressed the limitations and opportunities in the revised manuscript.

In the abstract and introduction, the authors motivate their study from a too macroscopic perspective of the problem. However, the limitation of the method seems to restrict the application of the method to idealized scenarios, whose ecological relevance is questionable. However, I believe that studying such idealized, small systems is an initial key step. I think the authors could try to discuss in more detail how such studies can help to understand the ecology of plankton systems.

We took the macroscopic perspective to motivate our study and put it in that context. While we still think this is relevant, we have followed the reviewer’s advice and de-emphasized the significance for large-scale processes by referring more to the microbial food web and less to the ocean food web, and we also refer to specific examples from larger organisms, where individual-level observations have furthered the understanding for higher-level dynamics to make it more clear how and when we envision that small-scale observations can provide mechanistic understanding with some relevance also for larger scale dynamics (see answer to R1.1 and changes in the manuscript file with highlighted changes).